# HOW MUCH CHAIN-OF-THOUGHT DO LLMS REALLY NEED FOR PHYSICS?

## ABSTRACT

Reasoning-focused language models are increasingly applied to AI for science, but evaluation has not kept pace: benchmarks largely measure end-task accuracy while ignoring whether models genuinely depend on their own reasoning traces. This gap is critical in domains like physics problem solving, where equations, units, and structured terminology make reasoning reliability both essential and testable. We introduce a systematic deletion framework that intercepts chain-of-thought (CoT) mid-generation, removes tokens, and measures downstream effects. Applied to three open-source models—Magistral, Phi-4, and Qwen-A3B—across multiple physics benchmarks, our method shows that models remain accurate under heavy deletions (40–60%) by "cramming" reconstructed steps into final answers. Overlap analyses reveal that deleted equations and facts often reappear, but inconsistently across strategies, exposing shallow and opportunistic reliance on CoT. These findings underscore that current accuracy-based evaluations are insufficient for scientific domains, and point toward the need for methods that assess reasoning faithfulness as a core requirement for advancing AI for science.

## 1 INTRODUCTION

Large language models (LLMs) are increasingly presented not only as generators of fluent text but as *reasoning systems*, capable of solving multi-step problems in mathematics, science, and beyond (Yao et al., 2023; OpenAI et al., 2024). A central technique behind this framing is *chain-of-thought* (CoT) prompting, which elicits step-by-step reasoning traces prior to a final answer (Wei et al., 2022a; Kojima et al., 2022). Yet a key question remains: do models genuinely *depend* on these traces, or do they function mainly as scaffolding for answer generation? While CoT has been argued to provide partial monitorability of internal processes (Korbak et al., 2025), evidence suggests limited dependence. Models can output correct answers while producing unfaithful reasoning traces (Turpin et al., 2023); correctness alone does not establish whether reasoning was used (Lanham et al., 2023); and in many cases, models regenerate plausible but unused intermediate steps (Lyu et al., 2023). This distinction is critical: faithfulness in CoT is not equivalent to interpretability or explainability (Barez et al., 2025), but rather concerns whether the scratchpad faithfully represents the computations that yield the final answer.

We investigate this faithfulness gap—and the broader evaluation gap of LLM reasoning—in the context of *physics problem solving*. While prior work has examined CoT faithfulness in general settings, its implications for *AI-for-Science* remain underexplored. Physics provides a stringent testbed: unlike open-ended reasoning tasks, it requires precise manipulation of equations, units, and numerical calculations, where small errors propagate into incorrect results (Shapira et al., 2023; Kosinski, 2024). At the same time, physics is central to visions of domain-specialized foundation models (Barman et al., 2025), making it both scientifically important and methodologically revealing. More broadly, physics exemplifies the reliability challenges facing *AI-for-Science*, where robust reasoning is essential for reproducibility, hypothesis generation, and discovery across disciplines (Bommasani et al., 2023; Stevens et al., 2023; Eger et al., 2025).

To this end, we evaluate three recent reasoning-oriented LLMs—Magistral (Rastogi et al., 2025), Phi-4 (Abdin et al., 2024), and Qwen-A3B (Qwen, 2025)—on three physics benchmarks of varied difficulty: Undergraduate Physics (Xu et al., 2025), PhyBench (Meng et al., 2024), and PhysReason (Zhang et al., 2025). Our study proceeds in three stages: (1) establishing baseline performance un-

Figure 1: Overview of Experiments. We study how LLMs consume chain-of-thought scratchpads in Physics problem solving. By manipulating the reasoning prompts and deleting intermediate steps, we evaluate accuracy, answer length, and reconstructions of missing steps in the final answer.

der direct and CoT prompting; (2) introducing a systematic deletion framework that intercepts CoT traces mid-generation and removes tokens before decoding; and (3) conducting a rigorous faithfulness analysis using information-overlap metrics and domain-aware matching to test whether deleted content reappears in final answers. Together, these steps provide a structured characterization of how open-source reasoning models use—or bypass—their CoT traces in scientific problem solving, exposing a reasoning-dependence gap that motivates new evaluation protocols and model designs emphasizing not only accuracy but also fidelity, with direct implications for AI-for-Science.

**In summary, our work introduces deletion-based probing as a new methodology for evaluating reasoning dependence in scientific domains**, and applies it to physics as a structured, high-stakes testbed. This framework yields both methodological advances and empirical insights into the limits of chain-of-thought reasoning.

1. **A systematic deletion framework** for probing reasoning dependence in LLMs. Our framework introduces a simple yet novel evaluation paradigm: intercepting CoT mid-generation, deleting intermediate tokens, and measuring their downstream impact on decoded information funneling and final answer quality.

2. **An empirical characterization of robustness and cramming**, showing that accuracy remains stable under moderate deletions (up to ∼40–60%) before collapsing, and that models exhibit compensatory "cramming" behavior—producing longer final answers that attempt to reconstruct missing reasoning.

3. **A rigorous faithfulness analysis** leveraging the structured nature of physics and mathematics. Using overlap metrics (Jaccard and Manhattan distance), we compare original CoT traces with regenerated reasoning across deletion sweeps. The domain's clear structure— equations, units, and terminology—enables precise quantification, revealing that models often reintroduce deleted content, producing surface-level agreement without genuine reasoning dependence.

These contributions highlight both the promise and the pitfalls of current reasoning models in scientific domains. They underscore the need for evaluations—and ultimately model designs—that prioritize *faithfulness* in reasoning, not just accuracy, with broader implications for AI-for-Science and structured problem solving.

## 2 PROBLEM SETUP

We systematically probe how LLMs use CoT reasoning in physics problem solving by actively intercepting and selectively deleting intermediate scratchpad prior to decoding. These CoT deletion experiments allow us to assess whether scratchpads are faithfully consumed, how models respond to partial removal of reasoning steps, and the extent to which missing information is reconstructed in the final outputs. An overview of our methods and evaluation metrics is presented in Figure 1.

## 2.1 TASKS AND DATASETS

We evaluate on three physics benchmarks of increasing difficulty: UG Physics (easiest), PhysReason (intermediate), and PhyBench (hardest). UG Physics emphasizes factual recall and straightforward applications of physics principles, while PhysReason combines knowledge-based and reasoning-intensive problems. PhyBench, the most challenging, requires advanced multi-step reasoning and deep conceptual understanding.

- **UG Physics:** Undergraduate-level problems in classical mechanics, electromagnetism, and thermodynamics, requiring multi-step reasoning and the application of standard formulas and units.
- **PhysReason:** A benchmark of 1,200 problems spanning factual recall (30%) and reasoning-based questions (70%), with varying difficulty.
- **PhyBench:** A Physics Olympiad-style benchmark designed to test complex reasoning, with problems requiring both deep conceptual insights and numerical problem solving.

## 2.2 MODELS

While a substantial body of recent work (Wei et al., 2022b;a; Nazi et al., 2025) on CoT prompting has focused on closed-source LLMs accessed through APIs (e.g., PaLM, LaMDA, GPT variants), such settings typically restrict visibility into intermediate reasoning traces and limit opportunities for controlled interventions. To enable a more systematic investigation, we instead turn to open-source reasoning LMs, which allow us to directly intercept the CoT scratchpad prior to decoding. This access enables us to precisely manipulate intermediate reasoning and study the effects of different types of CoT deletions. Concretely, we evaluate three open-source LLMs spanning distinct architectures and pretraining regimes:

- **Phi-4:** A 14B reasoning-focused model, fine-tuned on curated chain-of-thought prompts and reinforced via supervised and RL methods, excelling in mathematical and logical reasoning tasks.
- **Qwen-A3B:** A 30.5B general-purpose Mixture-of-Experts LLM with a four-stage training pipeline including chain-of-thought cold start, reasoning RL, and thinking-mode fusion, optimized for multi-step reasoning and long-context understanding.
- **Magistral:** A reasoning-focused model from Mistral AI, with the open-sourced *Small* variant (24B parameters) trained via a reinforcement learning pipeline (GRPO) to improve multi-step reasoning and instruction following, including multilingual chain-of-thought capabilities.

All models are prompted in reasoning mode (explicit CoT scratchpad), and sampled with nucleus sampling (temperature $T = 0.6$ to $0.7$, top-$p = 0.95$).

## 2.3 CALIBRATING CHAIN-OF-THOUGHT

**Reasoning explicitness and prompting style** To evaluate the role of reasoning in model performance, we vary the *prompting style*, which controls how much a model is encouraged to rely on CoT. We distinguish between two categories of prompts (see §D for the full templates):

1. **Full Reasoning:** The model is prompted to work through the problem in detail, producing a step-by-step derivation with comprehensive explanations of the relevant physics concepts and mathematical steps. The emphasis is on completeness, transparency of reasoning, and not skipping intermediate steps. (This corresponds to the *High Reasoning* setting.)

2. **Less Reasoning:** The model is encouraged to solve the problem with reduced deliberation. This includes two sub-levels:
   - *Medium Reasoning:* Reasoning is still step-by-step, but concise and focused, avoiding excessive elaboration.
   - *Low Reasoning:* The model is asked to minimize reasoning, providing a quick answer with only minimal or implicit thought steps.

This setup allows us to baseline the differences in model performance that arise from the inherent CoT reasoning reliance. We note that in most of our experiments beyond the initial comparison, we use the medium reasoning prompt by default.

**Number of Samples**   We calibrate the number of data points and runs sufficient for our experiments based on ablation studies.

## 2.4   METRICS AND EVALUATION

We quantify model behavior along three axes:

- **Score:** Evaluated with Claude-4 Sonnet as judge, scoring 0–1 based on correctness, derivation accuracy, logic, formatting, and clarity. The model compares each solution to the expected answer, penalizing deviations.
- **Final Answer Length:** Number of characters generated in the answer, used to detect cramming behavior.
- **Information Overlap:** Fraction of deleted CoT elements that reappear in the final answer, measured using Bag-of-Words metrics: Jaccard similarity and Manhattan distance.

This setup allows systematic evaluation of both the necessity and faithfulness of CoT reasoning in LLMs for physics problem solving.

## 3   EXPERIMENTAL RESULTS

We experiment with the role of CoT scratchpads in physics reasoning tasks, focusing on whether they are faithfully used, when they become essential, and how models compensate under manipulation. We evaluate three recent LLMs—Phi-4, Qwen-A3B and Magistral—on three physics benchmarks: UG Physics, Phy-Bench, and PhysReason. For all our experiments, we use nucleus sampling with temperature $T = 0.6$ to 0.7, top-$p = 0.95$.

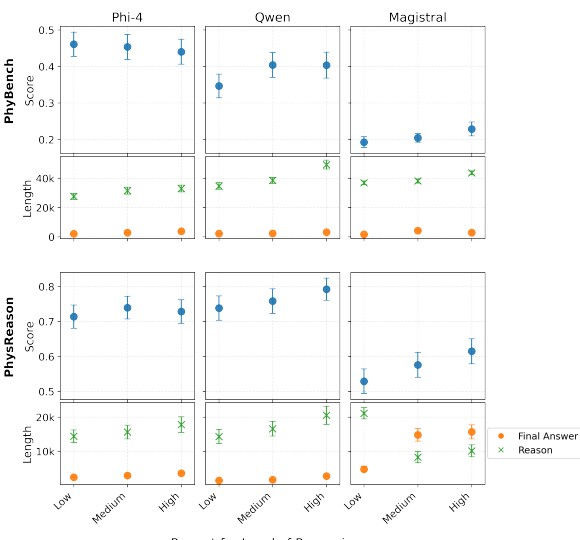

### 3.1   PROMPTING AND CALIBRATION

We begin by investigating whether explicit reasoning traces improve performance beyond direct answer generation.

**Reasoning explicitness and prompting.** We find a consistent trend across models and datasets: performance improves with the explicitness of reasoning. When prompted with *Full Reasoning*, models often achieve the highest accuracy, benefiting from detailed step-by-step derivations that enforce intermediate consistency checks (e.g., writing governing equations, performing algebraic transformations). Under the *Less Reasoning* settings, accuracy declines, reflecting that concise reasoning sketches, while still helpful,

Figure 2: Prompting styles evaluation across 2 datasets and 3 models. **Full Reasoning (High):** the model shows all intermediate steps before the final answer. **Less Reasoning (Low/Medium):** the model provides briefer reasoning. We observe that higher explicitness generally leads to better answer quality.

provide fewer opportunities for the model to correct errors in intermediate steps.

We evaluate results using Claude-4 Sonnet as a judge model, scoring each solution on a 0–1 scale based on correctness of the final answer, accuracy of the physics derivation, logical coherence, formatting, and clarity. The model is provided with the expected full answer for direct comparison, and large deviations are penalized. This evaluation confirms that higher reasoning explicitness consistently yields more reliable and logically coherent solutions.

Figure 2 summarizes these results by showing model performance across reasoning conditions; specifically, prompting models for more extensive reasoning (the *Full Reasoning* condition) yields higher judged derivation quality and greater solution coherence than prompts that elicit less reasoning.

**Calibration study.** To determine how many samples are required for stable estimates, we conduct a convergence analysis by increasing the number of independent prompt completions and computing the width of the confidence interval. Using bootstrapped results over 50 UG-Physics questions with 5 re-runs of the same data, we find that approximately *5 prompts* are sufficient to reduce the relative error bar below 10%. We also confirm this trend with quartile-based results, and adopt this setting as our standard calibration configuration in Figure 8.

## 3.2 CoT Deletion Sweeps

In §3.1, we confirm that longer, explicit CoT correlate with higher scoring solution, an unsurprising but important baseline. To probe how models rely on CoT during structured reasoning such as Physics, math or other AI for science related tasks, we conduct *systematic deletion experiments*. Figure 3 summarizes the effect of CoT deletion on model performance. Across all models and datasets, we observe that answer scores degrade when portions of the CoT are removed. In this figure, we focus specifically on physics-related annotations within the CoT, which we restrict to structured elements such as equations and units. We then compare two conditions: deleting all *annotated* (physics-structured) elements vs. deleting the remaining, *non-annotated* portions. In both cases, performance declines, but the removal of annotated facts produces a more detrimental effect on answer scores. We also observe that the final answer lengths sometimes slightly increases when reasoning with partially deleted CoT.

To better understand the slight increase in final answer length, we systematically characterize this effect. Specifically, we intercept the scratchpad and remove $k\%$ of CoT tokens ($k \in [0, 100]$) before the final answer. We compare three deletion strategies: (1) **from-the-end deletion**, truncating the last $k\%$ of tokens; (2) **random**

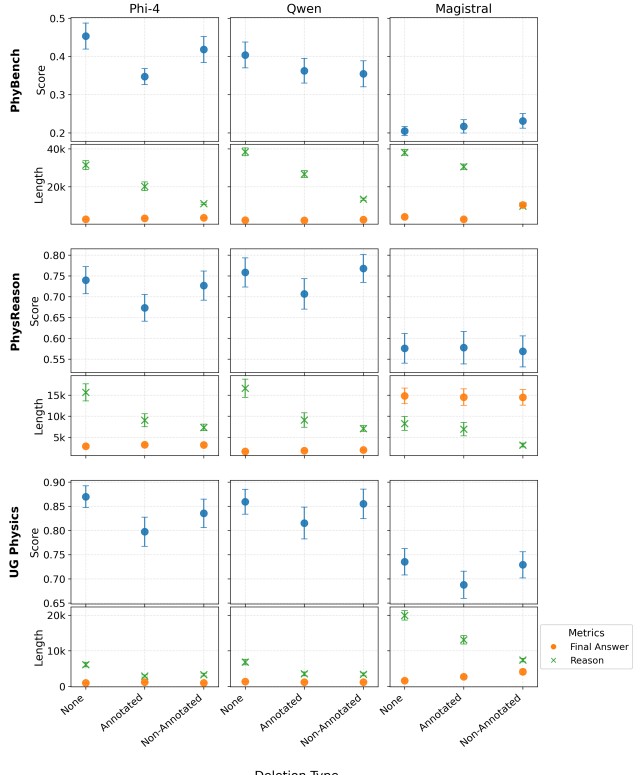

Figure 3: Effect of CoT deletions on physics benchmarks across models. **None** = full CoT, **Annotated** = deletion of physics-structured elements (e.g., equations/units), **Non-Annotated** = deletion of remaining content. Removing any portion lowers scores (blue dots), with annotated deletions most detrimental. The final answer length (orange dots, in character counts) slightly increases with CoT deletions.

**deletion**, removing tokens uniformly at random; and (3) **physics-aware deletion**, where another model (Claude-4 Sonnet) identifies physics-related tokens for removal. Across strategies, accuracy declines monotonically with greater deletion, while answer length increases. This possibly indicates that models attempt to *reconstruct lost reasoning* directly in the answer stage—a behavior we term **cramming**.

**From-the-end deletion sweep.** We delete $k\%$ of CoT tokens from the end, sweeping $k \in [0, 100]$. Accuracy remains stable until approximately $40\%$ deletion, after which it drops, as shown in figure 6. In general, we observe an X-shaped pattern in the answer length: as CoT reasoning is deleted, the final answer length steadily increases, compensating for the missing reasoning. Beyond roughly $40\%$ deletion, accuracy declines, though in some cases this is partially offset by a large increase in the final answer length, possibly indicated by a slight uptick in accuracy in panels b), c), and f) of the undergraduate physics results in figure 6.

**Random deletion sweep.** We randomly delete $k\%$ of CoT tokens, sweeping $k \in [0, 100]$. Accuracy remains stable until approximately $60\%$ deletion, after which it *drops sharply*. Despite slightly higher variance compared to from-the-end deletion, we observe the same X-shaped pattern: as reasoning is removed, the final answers become steadily longer, compensating for the missing CoT tokens. At high deletion levels, this effect is especially pronounced, with answers often becoming significantly longer. Figure 11 in §B illustrates this trend.

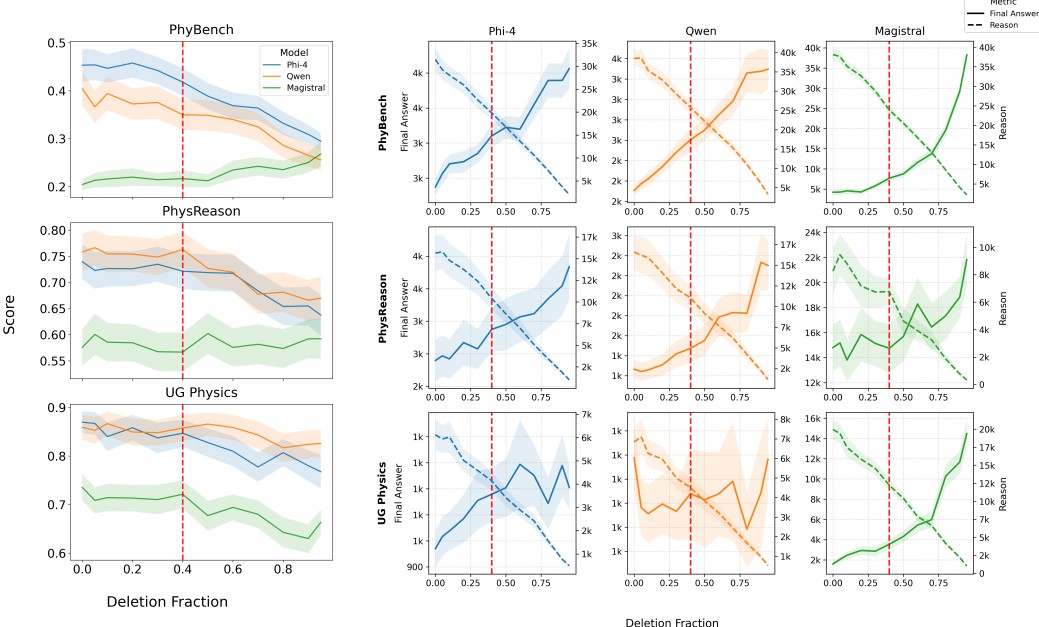

Figure 4: Final answer scores under end deletion. Accuracy begins to drop noticeably around $40\%$ deletion (red dotted line).

Figure 5: Final answer length under end deletion. As more reasoning is removed (dotted line), answers (solid line) tend to become longer.

Figure 6: From-the-end deletion-sweep visualizations.

**Physics-aware deletion.** We selectively remove domain-relevant content by tagging physics-specific spans (e.g., equations, constants, unit conversions) with Claude-4 Sonnet and deleting $k\%$ of these tokens. Accuracy declines steadily but less abruptly than in random or end deletion (Figure 14 in §C). Answer length, however, increases sharply once $70$–$80\%$ of annotated tokens are removed, indicating partial compensation until critical facts are lost. These results highlight the importance of domain-specific knowledge in maintaining reasoning fidelity.

# 4 ANALYSIS AND DISCUSSION

Our experiments reveal several robust patterns in how LLMs utilize chain-of-thought (CoT) scratch-pads for physics reasoning, which we analyze below.

## 4.1 CRAMMING BEHAVIOR

Across all three models and datasets, we observe a striking pattern: *when substantial portions of CoT are deleted, the final answer length increases sharply*, often with reconstructed equations or intermediate steps reappearing in the final output. We term this compensatory behavior **cramming**. While we do not probe internal mechanisms directly, these results suggest that LLMs may draw on internalized physics knowledge or learned solution templates to regenerate missing reasoning steps during answer decoding.

This behavior appears consistently across all three deletion strategies. For **end deletion**, Figure 6 shows that cramming emerges once roughly $40\%$ of the CoT is removed, followed by a gradual increase in final answer length. For **random deletion**, Figure 11 indicates that cramming becomes pronounced at around $60\%$ deletion, again with a steady length increase thereafter. Finally, under **physics-aware deletion**, Figure C shows a much more gradual decline in accuracy, with degradation only becoming noticeable at $70$–$80\%$ deletion. At this point, however, the model exhibits a sharp spike in final answer length, consistent with cramming behavior.

## 4.2 INFORMATION OVERLAP AND RECOVERY

Our analyses reveal a dual behavior in model reasoning under CoT deletion: while models often attempt to reconstruct missing structured information, the recovery is not guaranteed to be faithful, since the final answer score mostly does not recover across 3 different deletion strategies. In some cases (e.g., Phi-4 on undergraduate physics), models seem to substitute alternative reasoning rather than recovering the original, suggesting that reconstruction is heuristic and opportunistic rather than systematic.

To quantify this phenomenon, we measure whether deleted information reappears in final answers. Because physics reasoning relies heavily on structured content—such as specialized terminology, equations, and units—we evaluate recovery using strict token-overlap metrics between the generated answers and the original CoT before deletion. This allows us to assess both the degree of redundancy in model reasoning and the limits of faithful recovery across deletion sweeps.

**Defining overlap.**   We define **information overlap** as the intersection between (i) the original CoT prior to deletion and (ii) new content generated in the final answer across deletion sweeps.

**Quantification.**   We measure overlap using two complementary metrics:

1. **Lexical Overlap (Jaccard Similarity):** captures shared vocabulary, ignoring frequency. For passages $p_1$ and $p_2$, let $V(p)$ denote the set of unique tokens. Then

$$\text{Jaccard}(p_1, p_2) = \frac{|V(p_1) \cap V(p_2)|}{|V(p_1) \cup V(p_2)|}. \tag{1}$$

2. **Frequency Overlap (Manhattan Distance on Bag-of-Words):** captures distributional similarity in word usage. For passages $p_1, p_2$ with bag-of-words representations $\text{bow}(p_1), \text{bow}(p_2) \in \mathbb{R}^d$, where each dimension counts token frequency, we compute

$$D_{\text{Manhattan}}(p_1, p_2) = \sum_{i=1}^{d} \left| \text{bow}(p_1)_i - \text{bow}(p_2)_i \right|. \tag{2}$$

These metrics highlight different aspects of recovery: Jaccard similarity reflects vocabulary-level reuse, while Manhattan distance accounts for shifts in token frequency distributions.

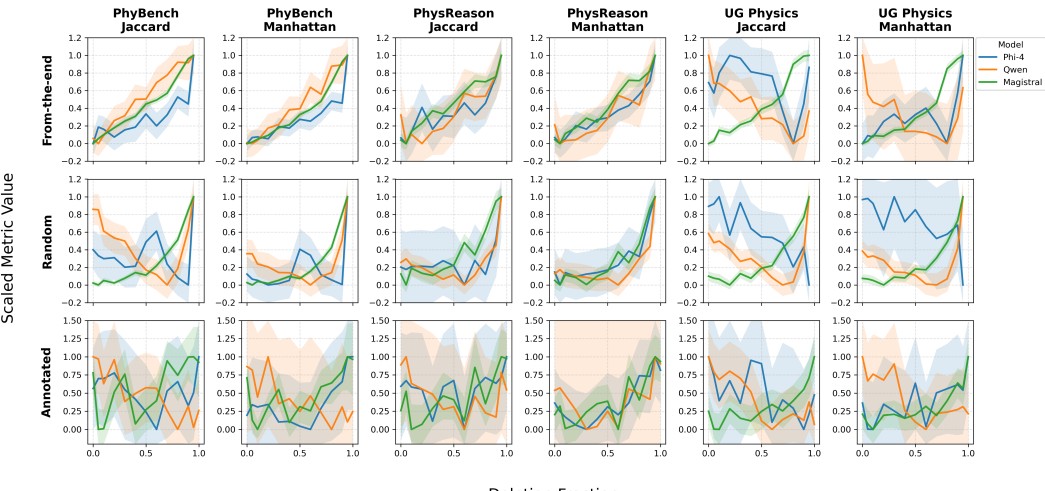

Figure 7: Information overlap under deletion sweeps. Each panel reports scaled overlap metrics (Jaccard similarity and Manhattan distance) between deleted CoT content and regenerated final answers, across three datasets (PhyBench, PhysReason, UG Physics) and three models (Phi-4, Qwen, Magistral). Rows correspond to deletion strategies (end, random, physics-aware). Overlap generally increases with deletion fraction, consistent with models attempting to reconstruct lost content. The effect is most systematic under end deletion, emerges later under random deletion, and appears noisier under physics-aware deletion. Shaded regions indicate the standard error.

**Findings.** Figure 7 shows that information overlap between deleted CoT spans and regenerated answers increases as deletion progresses, but the pattern varies across strategies and datasets. Under **end deletion**, overlap rises smoothly and consistently across all models and benchmarks, reflecting systematic attempts to reconstruct truncated reasoning. In contrast, **random deletion** yields delayed overlap growth (becoming pronounced only beyond ∼60% deletion) and exhibits higher variance, suggesting that scattered removals are harder to recover from. **Physics-aware deletion** produces the noisiest trends: overlap remains relatively flat until heavy deletion (70–80%), at which point sharp spikes appear, consistent with late-stage cramming. Across datasets, recovery is most stable on PhyBench and PhysReason, whereas UG Physics displays greater variability, with some models substituting alternative reasoning instead of reproducing the deleted content.

Taken together, these results suggest that while models opportunistically recover missing information, such recovery often reflects surface-level similarity rather than genuine fidelity to the original CoT. This points to a deeper conflict between CoT reasoning as written in the scratchpad and the model's own decoding process: reconstructed content may be heuristically generated rather than faithfully recovered, raising questions about the faithfulness of CoT traces as evidence of underlying reasoning.

## 4.3 IMPLICATIONS FOR COT FAITHFULNESS

Our findings provide new perspective on the *faithfulness* of chain-of-thought (CoT) reasoning. By faithfulness, we refer to the extent to which the scratchpad explicitly reflects the internal computations that lead to the model's final prediction, rather than merely serving as a plausible post hoc justification. Across deletion sweeps, we observe that: (i) not all intermediate steps in the scratchpad are faithfully required for correct answers, and (ii) models deploy compensatory mechanisms—such as cramming—to regenerate missing information directly in the final answer.

These observations suggest that CoT scratchpads are simultaneously *informative* and *redundant*. On one hand, they contain structured reasoning traces that improve fidelity when preserved. On the other hand, their partial bypassability raises the possibility that CoT text is not a transparent window into model reasoning, but rather an externalization that can diverge from the underlying decision process. For interpretability, this cautions against treating CoT explanations as fully faithful accounts. For

prompting and system design, it highlights the need to explore strategies that promote reliance on genuine intermediate reasoning rather than heuristic reconstruction.

These findings also carry practical implications. First, because models can often reconstruct missing information in the final answer, *early stopping of CoT generation* may provide a cost-effective way to save tokens without proportionally sacrificing accuracy. Second, the fact that useful information can be compressed and reconstructed suggests that prompting strategies could be redesigned to elicit more concise yet effective reasoning traces. In short, while CoT can illuminate aspects of model reasoning, it cannot yet be assumed to faithfully reveal it.

### 4.4 LIMITATIONS

Our study has several limitations. First, our experiments are scoped to physics reasoning tasks and three representative LLMs. While this domain is specialized, it is also representative of structured reasoning challenges central to AI-for-science more broadly, suggesting that the qualitative patterns we observe may generalize beyond physics. Second, our conclusions are drawn from *observable outputs*; we do not analyze latent representations, internal attention patterns, or decoding dynamics, which may reveal additional mechanisms of information recovery. Third, although deletion sweeps demonstrate consistent trends across datasets and models, further work is required to test their robustness across other reasoning domains (e.g., mathematics, commonsense) and architectures.

Future research should expand to diverse domains and model families, and probe the *mechanistic basis* of cramming and overlap behaviors—for example, whether they arise from memorized templates, latent redundancy in representations, or adaptive decoding strategies. Additionally, scaling studies could clarify whether larger models exhibit more faithful CoT usage or simply stronger compensatory reconstruction.

## 5 CONCLUSION

CoT scratchpads play a dual role in physics reasoning tasks central to AI for science: they boost accuracy when intact but can be bypassed through *cramming*, where models reconstruct missing steps in final answers. This shows CoT traces are both informative and redundant, raising concerns about their **faithfulness** as evidence of reasoning. For interpretability, CoT should not be treated as transparent explanations; for system design, they highlight opportunities to trade off efficiency and reasoning fidelity. Advancing AI for science will require evaluation methods that go beyond accuracy to enforce faithfulness, ensuring that intermediate steps genuinely reflect underlying computations.

## 6 RELATED WORKS

**Reasoning-Focused Models.** Recent LLMs increasingly incorporate reasoning-oriented instruction tuning and reinforcement learning to improve multi-step problem solving. Phi-4 (Abdin et al., 2024) is fine-tuned on curated chain-of-thought datasets and refined using reinforcement learning, achieving strong performance on mathematical, logical, and planning tasks despite its moderate parameter count. GLM-4.5-Air (Zeng et al., 2025) leverages a Mixture-of-Experts (MoE) architecture and multi-stage expert iteration with RL to support hybrid reasoning and agentic behaviors. Qwen-A3B (Qwen, 2025) uses a four-stage training pipeline combining reasoning RL, chain-of-thought cold-start, and thinking-mode fusion, optimizing multi-step reasoning and long-context comprehension.

**Chain-of-Thought Faithfulness.** While chain-of-thought prompting improves multi-step reasoning(Wei et al., 2022a;b; Yao et al., 2023), recent work highlights that generated reasoning steps may be unfaithful, containing errors or unsupported inferences (Barez et al., 2025). Faithfulness-focused approaches, including self-consistency decoding (Cheng et al., 2025; Wang et al., 2023) and verification-based RL fine-tuning(Su et al., 2025; Peng et al., 2025), aim to ensure that intermediate steps reliably lead to correct final answers. Models such as Phi-4, Qwen-A3B, and Magistral-Small incorporate elements of reasoning supervision and RL that may indirectly improve CoT faithfulness, although systematic evaluation of faithfulness remains an open challenge.

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

## A  CALIBRATION

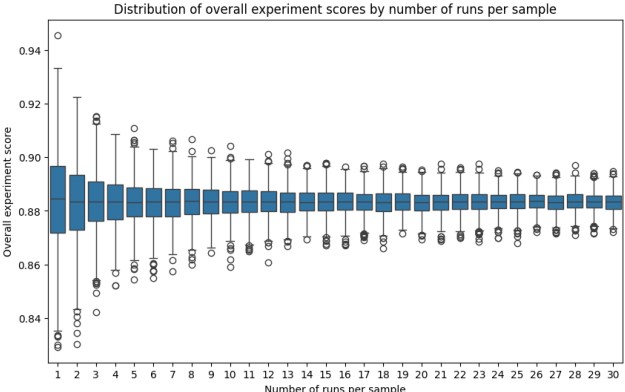

Figure 8: Calibration curve: error bar width vs. number of samples. Error stabilizes at around ∼5 samples.

## B  RANDOM DELETION SWEEPS

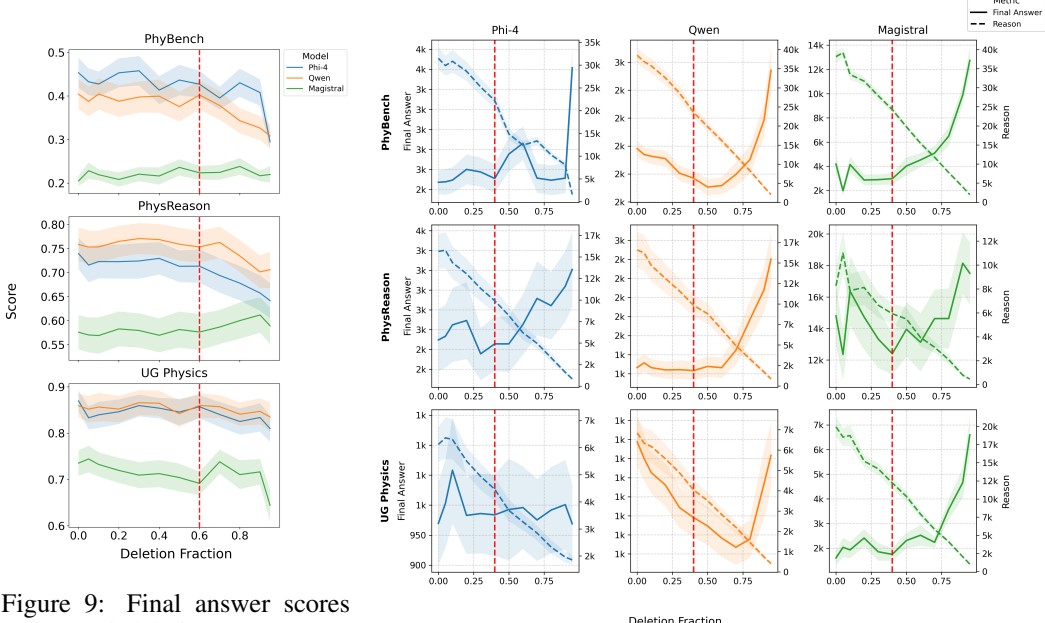

Figure 9: Final answer scores under end deletion. Accuracy begins to drop noticeably around 60% deletion (red dotted line).

Figure 10: Final answer length under end deletion. As more reasoning is removed (dotted line), answers (solid line) tend to become longer.

Figure 11: Effects of **random** deletion on model performance. Accuracy declines while answer length increases as larger portions of the chain of thought are truncated.

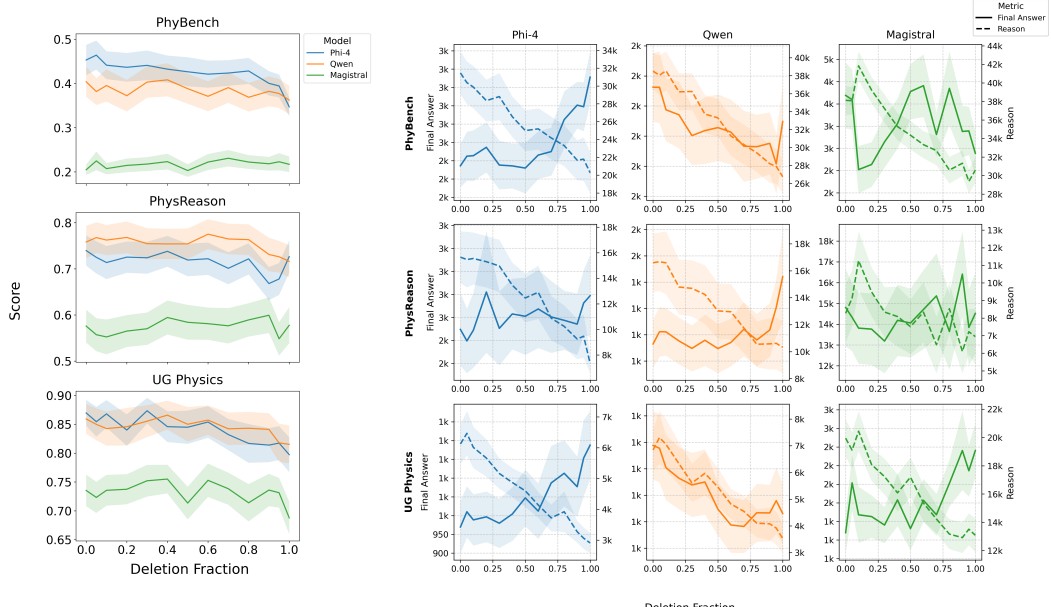

Figure 12: Final answer scores under physics-aware deletion. Score decreases gradually, with a less abrupt drop compared to other deletion methods.

Figure 13: Final answer length under physics-aware deletion. Answer length increases, particularly sharply when 70–80% of annotated physics tokens are removed.

Figure 14: Effects of physics-aware deletion on model performance. Accuracy declines steadily, while answer length increases sharply once most physics-related CoT tokens are removed.

## C  PHYSICS AWARE DELETION SWEEPS

## D  PROMPT TEMPLATES

We include the exact prompt templates used for each reasoning condition. All prompts were presented with the problem text substituted for {prompt}, and in some cases the expected final-answer instruction substituted for {final_answer_prompt}.

### D.1  HIGH REASONING (FULL REASONING)

```
{prompt}

Please solve this physics problem step by step. Be very thorough
    in your reasoning.

Think through the key physics concepts and mathematical steps
    needed. Do not skip any steps.

{final_answer_prompt}
```

### D.2  MEDIUM REASONING

```
{prompt}

Please solve this physics problem step by step. Be concise but
    thorough in your reasoning.
```

```
Think through the key physics concepts and mathematical steps
    needed, but keep your reasoning
focused and efficient. Avoid excessive elaboration on basic
    concepts.

{final_answer_prompt}
```

### D.3 LOW REASONING

```
{prompt}

Please think very briefly about this problem. Do not spend too
    much time thinking.
Please provide an answer as soon as you can.
```

