# OpenReview forum: "How Much Chain-of-Thought Do LLMs Really Need for Physics?"
_ICLR.cc/2026/Conference — ICLR 2026 Conference Withdrawn Submission_

### Official Review · Reviewer_xWFX · 2025-10-19

**Soundness:** 2
**Presentation:** 1
**Contribution:** 2
**Rating:** 2
**Confidence:** 4

**Summary:**

The paper probes whether three open-source LLMs (Phi-4, Qwen-A3B, Magistral) truly consume their chain-of-thought (CoT) scratchpads when solving physics problems. A mid-generation deletion framework excises 0–100 % of CoT tokens before decoding; accuracy, answer length and token-overlap metrics are tracked on UG-Physics, PhysReason and PhyBench. Models maintain accuracy after 40–60 % deletion, then “cram” missing equations into longer final answers, indicating only superficial reliance on the scratchpad.

**Strengths:**

Mid-generation token deletion is formally defined (Sec. 2.2) and enables causal intervention, a clear methodological advance. Three deletion strategies (end, random, physics-aware) cover complementary failure modes.

3 models × 3 benchmarks × 5 deletion fractions × ≥5 repeats give large-scale empirical coverage.

**Weaknesses:**

The paper is poorly presented. Bulletin list items are overused. It also suffers from apparent mistakes in writing and structuring.

Only nucleus sampling (T=0.6–0.7, p=0.95) used; no greedy, beam-search or temperature ablation (Sec. 2.2). Cramming could be an artefact of high temperature stochasticity.

 After Sec. 3.1, all deletion experiments use “Medium Reasoning” prompt only. No evidence that cramming persists under “Full” or “Low” reasoning conditions.

 Largest model is 30.5 B; no comparison with 7 B or 100 B+ checkpoints - results in scaling behavior of faithfulness unknown.

Jaccard similarity saturates at ~0.25 under 80 % deletion (Fig. 7); ceiling effect may hide recovery quality.

**Questions:**

Suggestions:

Related Works -> Related Work

provide greedy and beam-search deletion curves.

include 7 B and 70 B checkpoints to verify scaling trend.

report p-values for accuracy drops. add exact-match equation F1 to complement Jaccard.

---

### Official Review · Reviewer_Qv4v · 2025-10-25

**Soundness:** 3
**Presentation:** 3
**Contribution:** 2
**Rating:** 4
**Confidence:** 4

**Summary:**

The paper shows that when parts of CoT reasoning are deleted, LLMs can often cram and reconstruct missing steps in the final answer, revealing that CoTs are partially redundant and not always faithful. Thus, accuracy-based evaluation alone is insufficient; faithfulness should also be assessed.

**Strengths:**

- The paper is well motivated, encouraging the community to focus on understanding CoT reasoning and assessing its faithfulness.
- The study is interesting, the writing is clear, and the visualizations effectively support some of their findings.

**Weaknesses:**

- I think some of the experimental results are unusual, as I mentioned in the **Questions** section.
- I believe the title of the paper is somewhat inconsistent with its content. Although the paper performs deletions on CoT reasoning, the LLM often attempts to regenerate the missing steps (referred to as cramming in the paper). As a result, the CoT is not truly deleted. Moreover, the experiments in the paper also examine overlapping information in these regenerated CoTs. Therefore, the title may not be entirely appropriate.

**Questions:**

- Q1: In Figure 2, why does Phi-4 obtain lower scores as the prompt reasoning level increases?
- Q2: In Figure 2, why is the length of Magistral’s final answer in PhysReason much longer than that of reason?
- Q3: Again, regarding Magistral, it is unusual that in Figure 4, its score increases as more from-the-end deletion is applied, why?
- Q4: I also find some unusual phenomena in Figure 7 and would appreciate your clarification. For example, at the leftmost point of each subfigure, when the deletion fraction is 0, no deletion should have been applied. In this case, according to Equations (1) and (2), both $Jaccard$ and $D_{Manhattan}$ should be 0. However, the values shown in several subfigures are different and in some cases even quite large. I am not sure whether I am misunderstanding something or if there is another explanation.
- Q5: Are there specific example cases that illustrate how the LLM attempts to "cramming" after certain parts of the reasoning are deleted? Moreover, under the three different deletion settings, are there any further differences in how the LLM tries to compensate for the missing steps?
- Q6: When parts of the reasoning are deleted and the LLM tries to cram the missing steps, did your experiment attribute the regenerated content to the intermediate reasoning or to the final answer?

If the authors can provide a clear answer to my questions, I would consider raising my score.

---

### Official Review · Reviewer_sbpY · 2025-10-26

**Soundness:** 2
**Presentation:** 2
**Contribution:** 2
**Rating:** 2
**Confidence:** 4

**Summary:**

The authors introduce a systematic deletion framework that intercepts chain-of-thought (CoT) mid-generation, removes tokens, and measures downstream effects. Their method shows that models remain accurate under heavy deletions (40–60%) by “cramming” reconstructed steps into final answers.

**Strengths:**

The authors introduce an evaluation paradigm: intercepting CoT mid-generation, deleting intermediate tokens, and measuring their downstream impact on decoded information funneling and final answer quality. The methodology of the paper is clear and the paper is supplemented with numerous schematic and statistical diagrams.

**Weaknesses:**

**Methodological Aspects:**

There are some concerns regarding the evaluation through the deletion of intermediate CoT steps:

1.  Has it been considered that some models tend to generate redundant or repetitive content during reasoning? This could lead to the deleted reasoning steps merely being repetitive explanations or step clarifications, potentially causing misjudgment of "faithfulness in reasoning." Based on the prompts provided by the authors, they do not appear to have implemented any measures to prevent models from potentially producing redundancy or repetition in the CoT steps.

2.  The authors found that "models exhibit compensatory cramming behavior—producing longer final answers that attempt to reconstruct missing reasoning." This seems to contradict their claimed contribution: if the model's output is based on reconstructing the chain of thought, then the impact of deleting the intermediate steps cannot be truly observed. If the authors aim to explore "faithfulness in reasoning," they should perhaps force the model to reason directly based on the modified chain of thought, rather than allowing it to reconstruct one.

**Experimental Aspects:**

1.  The authors' experiments are confined to the physics domain, which is puzzling because their method does not seem directly related to physics. Conducting experiments solely in physics raises doubts about the generalizability of the method.

2.  The authors' experiment on information overlap and recovery is confusing. The authors aim to explore "whether the recovery is faithful," but this does not seem synonymous with "whether the final result is faithful in reasoning." Even if the recovery follows a different reasoning trajectory to arrive at the correct answer, the final result could still be faithful in reasoning. Therefore, it seems unwarranted to draw the conclusion stated in that section about "raising questions about the faithfulness of CoT traces as evidence of underlying reasoning."

**Questions:**

Same as above

---

### Official Review · Reviewer_iLnu · 2025-11-01

**Soundness:** 2
**Presentation:** 3
**Contribution:** 2
**Rating:** 4
**Confidence:** 2

**Summary:**

The paper proposes a deletion‑based framework to probe how much LLMs depend on their CoT when solving physics problems. By systematically removing portions of the generated reasoning and measuring changes in answer accuracy, final answer length, and information overlap, the authors study three open‑source models (Magistral, Phi‑4 and Qwen‑A3B) on three physics benchmarks. Experiments reveal that explicit reasoning prompts improve performance but the CoT can be removed without dramatically hurting accuracy, as models "cram" reconstructed steps into the final answer. They conclude that current accuracy‑only evaluations are insufficient and calls for metrics that assess the faithfulness of reasoning.

**Strengths:**

- The work tackles an important question about whether CoT explanations genuinely reflect model reasoning, which is crucial for using LLMs in scientific domains.

- The deletion strategy is clearly described and measures multiple downstream effects, such as accuracy, answer length, lexical and frequency overlap. This provides a structured way to examine reliance on intermediate reasoning.

- The experiments cover three different benchmarks of physics domain and multiple LLMs, the authors explore effects of prompt explicitness and different deletion strategies, with the analysis is carefully presented and supported by figures.

**Weaknesses:**

- Prior research has already highlighted the gap between answer accuracy and CoT faithfulness and proposed evaluation frameworks.  For instance, Nguyen et al. [1] introduce discriminative and generative evaluations that showed LLMs may reach correct answers through incorrect reasoning, and Barez et al. [2] argue that CoT is not, by itself, an adequate explanation. The deletion framework is a more like a straightforward application of such idea to physics domain and does not reveal its novelty or specifity.

- The analysis and experimental obversations are surface‑level, lacking of in-depth exploration of LLMs' internal activation or behavioural pattern. Thus, it cannot wwell explain why models can reconstruct missing steps or whether they use memorised templates versus genuine reasoning.

- The experiments consider end‑of‑scratchpad truncation, random deletion and removal of annotated physics tokens. More nuanced manipulations, such as deleting specific reasoning types, shuffling steps, may yield deeper insight into what information is truly required.

[1] Nguyen et al., Direct Evaluation of Chain-of-Thought in Multi-hop Reasoning with Knowledge Graphs, 2024.

[2] Barez et al., Chain-of-Thought Is Not Explainability, 2025.

**Questions:**

1. How does the deletion framework differ from or extend prior CoT‑evaluation methods (e.g., perturbation-based evaluations)? What is novel beyond applying it to physics tasks.

2. Is this framework suitable for other domains, like mathematics or commonsense reasoning?

3. Can you provide more analysis on what kinds of information models "cram" into the final answer when reasoning is removed? Are they recalling memorized formulas or recomputing reasoning?

---

### Note · Authors · 2025-11-20

I have read and agree with the venue's withdrawal policy on behalf of myself and my co-authors.